# Variation of B cell subsets with age in healthy Malawians

Wilson L. Mandala[ID][1,2]*, Herbert Longwe[2,3]

1 Academy of Medical Sciences, Malawi University of Science and Technology (MUST), Thyolo, Malawi,
2 The Malawi-Liverpool-Wellcome Trust Clinical Research Programme, Blantyre, Malawi, 3 ICAP at
Columbia University in South Africa, Pretoria, South Africa

* wmandala2002@gmail.com

## Abstract

Although a number of previous studies have shown that different lymphocyte subsets, including B cells, vary with age, how different B cell subsets vary with age in Malawian population has not been shown before. We recruited Malawian participants of different ages and analyzed their venous blood samples for different B cell subsets. We found that both percentage and absolute counts of B cells varied with age peaking in the 7 to 12 months age group. Proportion of naïve B cells was highest in neonates and decreased with age whereas the percentage of memory B cells was lowest in neonates and increased with age. When we zeroed in on the age band within which the proportion of B cells was highest, both classical and activated memory B cells increased with age and the naïve followed the opposite trend. These results provide additional knowledge in our understanding of the dynamics of B cell subsets in individuals of a specific ethnicity as they age.

## Introduction

Several factors are known to affect various immunohaematological parameters in an individual including age, genetics, sex, altitude [1, 2] and social habits such as smoking and dietary patterns [3]. Most of these factors vary depending on the population and geographical area studied [3]. This has the implication that the reference values that have been established and validated for non-adults from one area cannot be used for adult patients in the same area neither can reference values established for a particular ethnic population reliably be used in interpreting haematological parameters of a different ethnic population [2].

For years now flow cytometric analyses have been in use for the diagnosis of various immune deficiencies [4]. The advent of the human immunodeficiency virus (HIV) in the late eighties resulted in the widespread use of flow cytometric analyses not only for diagnosis but also for the monitoring of HIV infection and its progression and also for other infectious diseases, immunologic disorders and malignancies [4–8] and also for establishing reference ranges for leucocyte or lymphocyte subsets based on age, gender or ethnicity in healthy individuals [1–3, 9, 10].

B cells are a part of the cell-mediated immune system which are mostly known for their role in the production of immunoglobulins that are a crucial component of protective

**Funding:** The study of the first cohort was funded by a PhD fellowship for Wilson Mandala from the Bill and Melinda Gates Foundation under the Gates

Malaria Partnership (GMP) for African countries whereas the second cohort study was funded by the Wellcome Trust Grant to the Malawi Government under the Health Research Capacity Strengthening Initiative (HRCSI) for which both Wilson Mandala and Herbert Longwe were beneficiaries.

**Competing interests:** The authors have declared that no competing interests exist.

immunity to infections [11]. Back in the nineties some researchers discovered that they could distinguish two populations of B cells in the human tonsils and peripheral blood (PB) through expression of CD27 surface antigen [12]. Since then CD27 has become an important marker of human memory B cells. CD27 expression on B cells increases gradually with age and cord blood B cells do not express this marker whereas about 40 percent of adults peripheral blood B cells are $CD27^+$ [13]. Subsequently our knowledge of B cell subsets has improved with the advances made in Flow Cytometry. It is now known that B cells in peripheral circulation are made up of about two thirds of naïve ($CD19^+CD21^{hi}CD27^-CD10^-$) which express either switched or unswitched the antibody isotypes, IgG, IgE and IgA, and one-third memory B cells which in turn express switched or unswitched IgM and IgD [11, 14].

More importantly, it is now well established that long lived protective humoral immune response depends on generation of memory B cells that are further subdivided into classical memory ($CD19^+CD27^+CD21^{hi}CD10^-$) and activated memory ($CD19^+CD27^+CD21^{lo}CD10^-$) B cells [15]. Transitional immature B cells area minor population of B cells in peripheral circulations that express an immature phenotype ($CD19^+CD10^+CD27^-CD21^{lo}$) and are less likely to be activated [11]. In addition, a more unique memory B cell subpopulation has now been identified that expresses the surface markers $CD19^+CD27^-CD21^{lo}CD10^-$ and is defined by the expression of the inhibitory receptor Fc-receptor-like-4 (FCRL4) [16]. These atypical memory B cells are functionally distinct from the $CD27^+$ memory B cells and reportedly capable of expanding in individuals living in malaria endemic area [15].

Previous reports have shown that leucocyte subsets and lymphocyte subset patterns are mainly affected by ethnicity, gender and environment factors [8, 10]. Thus, the interpretation of the cell-mediated immunity that has been affected by or is responding to an infectious or non-infectious disease is predominantly dependent on having the appropriate normal reference values. We thus conducted this study, firstly to establish normal ranges of absolute counts and percentages (as percentages of total lymphocyte counts) of B cells and their various subsets in healthy Malawians from birth to adulthood. Secondly, having established that the main changes in B cell subsets occur in children aged between 6 and 18 months, we conducted additional set of experiments to determine how different B cell subsets vary with age within this age group.

## Materials and methods

### Participants

For the first part of the study, participants, ranging from newborn babies to adults over 60 years old were recruited from 26th September 2006 up to 15th January 2007 when they came to Ndirande Health Centre for routine health checks, for vaccinations or to give birth. The adults included mothers of the new-born babies whose blood samples were collected on the day of giving birth. Informed written consent was obtained from each adult study participant. In the case of minors, written informed consent was obtained from the parent or guardian of every child. All participants were considered healthy if they had no active disease and had no fever and were not on any medication at the time of recruitment. Blood samples of all participants were screened for HIV and tested for malaria parasitaemia as previously reported [9] and the data of those found to be HIV-infected or parasitaemic were excluded from the final data analysis. Results of this study showed that, unlike other lymphocyte subsets, the proportion of B cell subsets expanded substantially between the ages of 6 and 18 months (S3 Fig in S1 File.). We therefore went further and investigated how the different B cell subsets varied within this age range.

Thus for the second part of the study, a longitudinal approach was adopted. Study participants were confirmed HIV negative children who were known to have been born from HIV negative mothers. Since the second part of the study involved recruitment of minors, informed written consent was obtained from each parent or guardian of every child who met all study eligibility criteria. The children were recruited between August 2012 to February 2014 at aged 6 months then followed up at 12-months stage and then at 18-months stage. At each stage an EDTA blood sample was collected and this was used for HIV tests, malaria parasitaemia determination, Full Blood Count (FBC) and for other immunological assays. During all these visits, the children were seen in the study clinic where they underwent a detailed physical examination.

For both studies, sample size calculation was done based on previous studies' observation aimed at determining different leucocyte subsets in healthy individuals. These studies had compared sets of fewer than 50 study participants per each age group and identified significant differences in some parameters [1, 2]. Therefore for this study, in order to examine differences between different groups, we planned to recruit 50 participants (25 male and 25 female) for each group in order to have 90% power to detect a difference in mean of 1s, where s is the pooled standard deviation and the 1% significance level is used (to adjust for multiple tests). This calculation is based on use of a Mann-Whitney test and assumes the data are uniformly distributed. Unfortunately, we were not able to achieve these intended sample sizes for both genders and ended up recruiting slightly more of one gender than planned. For the second part of the study the two limiting factors to attaining the intended samples sizes were finding children of the right age at the start of the study (6 months) and the willingness of the mothers to return during the two follow-up stages (12 and 18 months respectively).

## Blood sample analyses

For both the first and the second part of the study, EDTA blood samples were used for full blood counts and for Flow Cytometric analysis. In the case of new born babies in the first part of the study, cord blood was collected from the placenta immediately post-delivery. For the first part of the study, the study participants were grouped into twelve categories based on their ages. Immunophenotyping of blood samples by flow cytometry (Four Colour BD FACSCalibur) was performed as previously reported [9]. Briefly, 25 μL EDTA whole blood sample was labeled with the following monoclonal antibodies (mAbs) in a flow tube: CD27-phycoerythrin (PerCP), and CD19-APC the details of which are provided in S1 Table in S1 File. A separate tube was included which had a similar volume of whole blood sample but was labeled with the following isotype control mAbs: Anti-moGI-FITC, Anti-moGI-PE, Anti-moGI-PerCP and Anti-moGI-APC. The details of these isotype control antibodies are also provided in S1 Table in S1 File.

Red cells were then lysed by using FACSlysing solution (Becton Dickinson) and samples washed with PBS. Sample cells were acquired by using a Four-Colour FACSCalibur flow cytometer (Becton Dickinson). Routine calibration and internal quality assurance of the instrument was performed by using Calibrite Beads and FACSComp software (both Becton Dickinson). Sample data were analyzed by using CellQuest software (Becton Dickinson). Total lymphocytes were identified by light scatter characteristics and the following lymphocyte subpopulations identified as percentages of the lymphocyte gate: B cells (CD19+), memory B cells (CD19+CD27+) and naïve B cells (CD19+CD27-) as depicted in the Gating strategy in S1 Fig in S1 File. Proportion of B cells, memory B cells and naïve B cells were determined from subpopulation percentages whereas the absolute counts of the same cell subsets were calculated from the flow readout percentages and total lymphocyte counts from the hematologic analyzer.

For the second part of the study, 25 μL of the EDTA whole blood sample was incubated with anti CD19 APC, anti CD21 PE-cy5 (all from BD Pharmingen, San Jose, California), anti CD10 FITC and anti CD27 PE (eBiosciences, San Diego, California) details of which are provided in S1 Table in S1 File. Labeled blood samples were then lysed, washed as explained for the first group and analysed by flow cytometry (Nine colour Cyan (Beckman Coulter) and the gating strategy shown in S2 Fig in S1 File was used for classifying different B cell subsets. B cells were identified as $CD19^+$ and subpopulations of naïve as $CD19^+CD21^{hi}CD27^-CD10^-$, classical memory as $CD19^+CD27^+CD21^{hi}CD10^-$, activated memory as $CD19^+CD27^+CD21^{lo}CD10^-$, atypical memory $CD19^+CD27^-CD21^{lo}CD10^-$ and immature transitional as $CD19^+CD10^+CD27^-CD21^{lo}$.

## Statistical analysis

For the first part of the study, the statistical analysis involved dividing the study participants into 12 age groups as indicated in Table 3. Kruskal-Wallis Test was used to assess if there were statistically significant differences in the medians of B cell subsets' counts or proportions amongst the various age groups. Considering that comparisons for more than two groups were made, between-age-group comparisons of the different B cell subsets were assessed using Dunn's multiple comparison test and where a $p$ value of $<0.0125$ was obtained, the difference between specific age groups was considered statistically significant.

Analysis of variance was conducted using Statistical Package for Social Sciences (SPSS), version 14 (Norusis, SPSS, Chicago, IL, USA). The results in Table 3 are presented as medians and $10^{th}$ and $90^{th}$ percentiles of the absolute counts and percentage of B cell subsets population analysed after age stratification. Since we observed that there were no significant differences between male and female participants for any of the various B cell subsets (absolute counts or proportions) (S2 Table in S1 File) therefore data for male and female participants in each age group were combined and analysed together. For the box whisker plots in Fig 1, the top, bottom and line through the middle of the box correspond to the $75^{th}$, $25^{th}$ and $50^{th}$ percentiles (median), respectively. The whiskers extend from $10^{th}$ percentile at the bottom to $90^{th}$ percentile at the top. Outliers are presented as filled circles.

For the second part of the study, GraphPad Prism 6 (GraphPad, California, USA) and STATA (StataCorp, Texas, USA) were used for carrying out the statistical analysis and constructing the graphical presentation of the results. Kruskal-Wallis Test was used to assess if there were statistically significant differences in the medians of B cell subsets' counts or proportions between the three time points. Considering that comparisons for three time points were made, comparisons of the different B cell subsets between the set time points were assessed using Dunn's multiple comparison test and where a $p$ value of $<0.0125$ was obtained, the difference between specific time points was considered statistically significant.

## Ethics approval

Both the first and the second part of the study were reviewed and approved by the College of Medicine Research Ethics Committee (COMREC) and were assigned the following Protocol Numbers P.01/02/176 and P.05/10/954 respectively. Individual written informed consent was obtained from either the participating adults or from the parents or guardians of all the children who participated in the study.

## Results

For the first part of the study, a total of 715 participants were recruited, and these ranged from neonates to adults over 60 years old (Table 1). Slightly more female participants [399 (56%)]

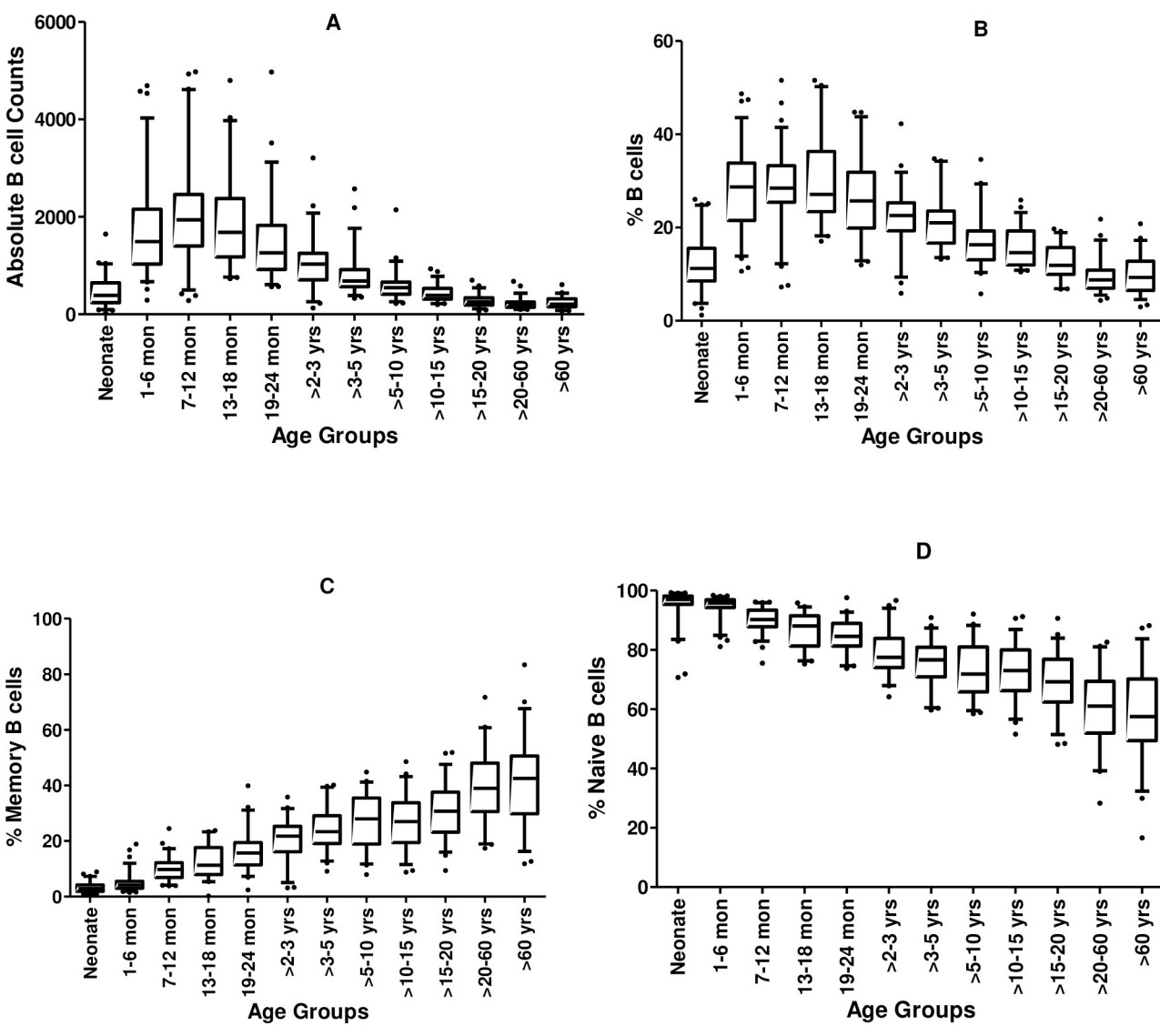

**Fig 1. Medians and 10<sup>th</sup> and 90<sup>th</sup> percentiles of various subsets of B cells in different age groups.** (A) Total B cells expressed as absolute cell counts, (B) Total B cells expressed as percentage of total lymphocytes, (C) Percentage of memory B cells (calculated from total B cells) and (D) Percentage of naïve B cells (calculated from total B cells).

took part in the study than male participants [316 (44%)]. The 55 participants in the group aged 60 years or more had a media age of 66 years with the age ranging from 60 to 92 years.

There were no significant differences between male and female participants for the frequency and absolute counts of various B cell subsets (S2 Table in S1 File). As such we combined data for female and male participants for each age range for subsequent statistical analyses for the first part of the study. Medians, 10<sup>th</sup> and 90<sup>th</sup> percentiles of the data for B cell counts and percentages are presented in Table 2 and in Fig 1.

Absolute B cell counts (Table 2 and Fig 1A) were low (387 cells/µl) in neonates, increased to the highest value (1,937 cells/µl) in the 7-to-12-moths age group, followed by a decrease reaching the lowest value (194 cells/µl) in the 20 to 60 years age group. A similar trend was observed when the B cells were presented as a percentage of the total lymphocytes (Table 2 and Fig 1B). The proportion of B cells was low in neonates (11.24%), increased gradually with age reaching

**Table 1. Age distribution of HIV-negative study participants according to gender.**

| Group Number | Age Range | Number of Female Participants | Number of Male Participants | Totals (Female and Male) |
|---|---|---|---|---|
| 1 | Neonates | 37 | 24 | 61 |
| 2 | 1 to 6 months | 44 | 24 | 68 |
| 3 | 7 to 12 months | 31 | 35 | 66 |
| 4 | 13 to 18 months | 15 | 27 | 42 |
| 5 | 19 to 24 months | 16 | 33 | 49 |
| 6 | >2 to 3 years | 20 | 28 | 48 |
| 7 | >3 to 5 years | 22 | 29 | 51 |
| 8 | >5 to 10 years | 30 | 26 | 56 |
| 9 | >10 to 15 years | 36 | 17 | 53 |
| 10 | >15 to 20 years | 29 | 29 | 58 |
| 11 | >20 to 60 years | 33 | 20 | 53 |
| 12 | >60 years | 31 | 24 | 55 |
| | | **399** | **316** | **715** |

the highest percentage in 7-to-12 months age group (28.46%) followed by a gradual decrease with the lowest percentage (9.28%) observed in the age group of 60 years and above.

The percentage of memory B cells were low in neonates (2.84%) and increased gradually with age with the maximum percentage (42.50%) observed in those aged 60 years and above (Table 2 and Fig 1C). An opposite trend was observed for the percentage of naïve B cells (Table 2 and Fig 1D). These were highest in neonates (97.03%) and decreased gradually with age with the lowest median percentage (57.50%) observed in those aged 60 years and above.

Having noticed that the main changes in the proportion and absolute counts of the B cells occur between 6 and 18 months (Fig 1A and 1B and S3 Fig in S1 File), we then focussed on this age band and adopted a longitudinal approach in order to identify the specific changes that occur to a few more subsets of B cells.

Just as was the case with the first cohort of the study, the percentage of B cells (as a percentage of total lymphocytes) did not significantly differ between 6, 12 and 18 months points (Table 3, Fig 2A). However the percentage of naïve B cells (as a percentage of total B cells) was

**Table 2. Medians and 10th and 90th percentiles of absolute B cell counts per μl of blood and for proportion of B cells, memory and naïve B cells in healthy participants of different ages.**

| Age | B Cell Counts/μL | B cells (%) | Memory B cells (%) | Naïve B cells (%) |
|---|---|---|---|---|
| Neonates (n = 61) | 387 (147–899) | 11.24 (4.86–19.40) | 2.84 (1.16–5.56) | 97.03 (91.27–98.84) |
| 1–6 months (n = 68) | 1,494 (739–2,882) | 28.71 (16.28–38.10) | 4.16 (2.52–8.59) | 95.83 (91.25–97.41) |
| 7–12 months (n = 66) | 1,937 (988–3,130) | 28.46 (17.60–36.92) | 9.75 (4.97–16.45) | 90.25 (83.90–95.03) |
| 13–18 months (n = 42) | 1,682 (931–3,431) | 27.11 (20.63–46.10) | 11.30 (5.92–20.64) | 88.06 (77.84–93.12) |
| 19–24 mon (n = 49) | 1,261 (701–2,382) | 25.70 (15.80–39.24) | 15.67 (9.44–25.40) | 84.56 (75.66–90.56) |
| >2–3 years (n = 48) | 1,030 (457–1,558) | 22.57 (13.23–27.43) | 21.72 (10.83–28.37) | 77.51 (69.60–89.17) |
| >3–5 years (n = 51) | 686 (433–1,393) | 21.00 (15.43–31.46) | 23.39 (14.22–34.66) | 76.61 (65.34–85.78) |
| >5–10 years (n = 56) | 545 (304–857) | 16.30 (11.32–25.50) | 27.98 (14.26–40.40) | 71.84 (61.38–85.62) |
| >10–15 years (n = 53) | 388 (241–672) | 14.60 (10.90–21.64) | 27.00 (15.68–39.60) | 73.00 (60.40–84.18) |
| >15–20 years (n = 58) | 253 (144–454) | 11.88 (7.51–18.23) | 30.72 (17.55–45.36) | 69.28 (54.11–82.45) |
| >20–60 years (n = 53) | 194 (109–293) | 8.79 (5.86–13.89) | 38.97 (25.52–56.25) | 61.03 (43.75–74.48) |
| >60 years (n = 55) | 208 (122–421) | 9.28 (5.35–15.53) | 42.50 (21.69–63.22) | 57.50 (36.78–78.31) |

The B lymphocyte subset showed a similar trend for both percentages and absolute counts.

**Table 3. Median proportion of B cell subsets in the study participants at different age points (6, 12 and 18 months).**

| Time Points | Proportion of CD19+ B cells subsets (%) | | | | | |
|---|---|---|---|---|---|---|
| | Total CD19+ | Naïve | Classical memory | Activated memory | Atypical memory | Immature transitional |
| 6-months (n = 29) | 20.7 (16.8–29.1) | 83.2 (80.7–87.9) | 5.26 (4.59–7.85) | 1.37 (0.85–2.16) | 4.21 (2.75–5.65) | 3.78 (2.21–5.09) |
| 12-months (n = 29) | 20.9 (15.2–27.4) | 73.6 (70.6–81.9) | 12.3 (8.68–14.4) | 2.77 (1.64–3.78) | 5.86 (3.84–7.77) | 3.38 (1.70–4.54) |
| 18-months (n = 29) | 18.8 (12.6–27.3) | 71.5 (63.2–74.0) | 13.0 (10.2–15.2) | 4.0 (3.17–5.22) | 7.53 (4.28–11.6) | 4.39 (2.41–5.47) |

significantly higher at 6 months stage (83.2%) compared to that at 12 months (73.6%) and 18 months (71.5%) (Table 3, Fig 2B).

As expected, the percentage of classical memory B cells was significantly lower at 6 months (5.26%) compared to 12 months (12.3%) and at 18 months (13.0%) (Table 3, Fig 2C) and a similar trend was observed for activated memory B cells (Fig 2D). There were no differences in the proportion of immature transitional B cells between any of the three age stages (Fig 2E) whereas the percentage of atypical memory B cells was significantly higher at 18-months age point than at 6-months point (Fig 2F).

## Discussion

B cells play a fundamental role in human immune response against infections since, once they differentiate to plasma B cells, they serve as antibody producing cells. Thus establishing how these cells are affected by age in healthy individuals is important as these values can then be used as a reference point in explaining any variation that might occur due to infection or vaccination [17–19].

Over the years various investigators have conducted studies with the aim of determining the normal B cell subset ranges for their population. Table 4 summarises the distribution of B cell subsets in study participants from various countries.

The results of our study compared well with the results of some similar studies conducted in other countries with almost identical values for B cell counts for participants in Tanzania and Germany (Table 4). Although we [28] and others [19] have previously shown that ethnicity does have an effect on some lymphocyte subsets, the results presented in Table 4 seem to suggest that, with the exception of the Cuban study [23], proportion and cell counts of B cell subsets do not differ substantially between the different ethnic groups. The higher proportion and absolute counts for B cells observed in the Cuban population was explained as a possible result of continuous exposure to viral and other infections [23].

CD27 is a marker of primed memory cells and its engagement promotes the differentiation of memory B cells into plasma cells [29]. Evaluation of maturation-specific phenotypes in peripheral blood is useful in providing evidence for prior antigen exposure and assessing the resulting lymphocytes homing patterns [30]. Previously some investigators [31] demonstrated an increase of CD27+ B cells and a decrease of CD27- B lymphocytes in elderly. These age-related changes were also observed in our study.

Recent studies on how B cell subsets vary with age [19, 32–34] have also shown similar trends to those observed in our study in terms of naïve B cells (decreasing with age) classical memory B cells and activated memory B cells (increasing with age). The observation made in our study showing significantly higher percentage of atypical memory B cells at 18 months compared to 6 months is fascinating and needs to be studied further.

Similar to other studies [21, 26, 35], we had previously shown that there were no significant differences between male and female participants for the frequency and absolute counts of various lymphocyte subsets including B cells [9] although the other studies had found some

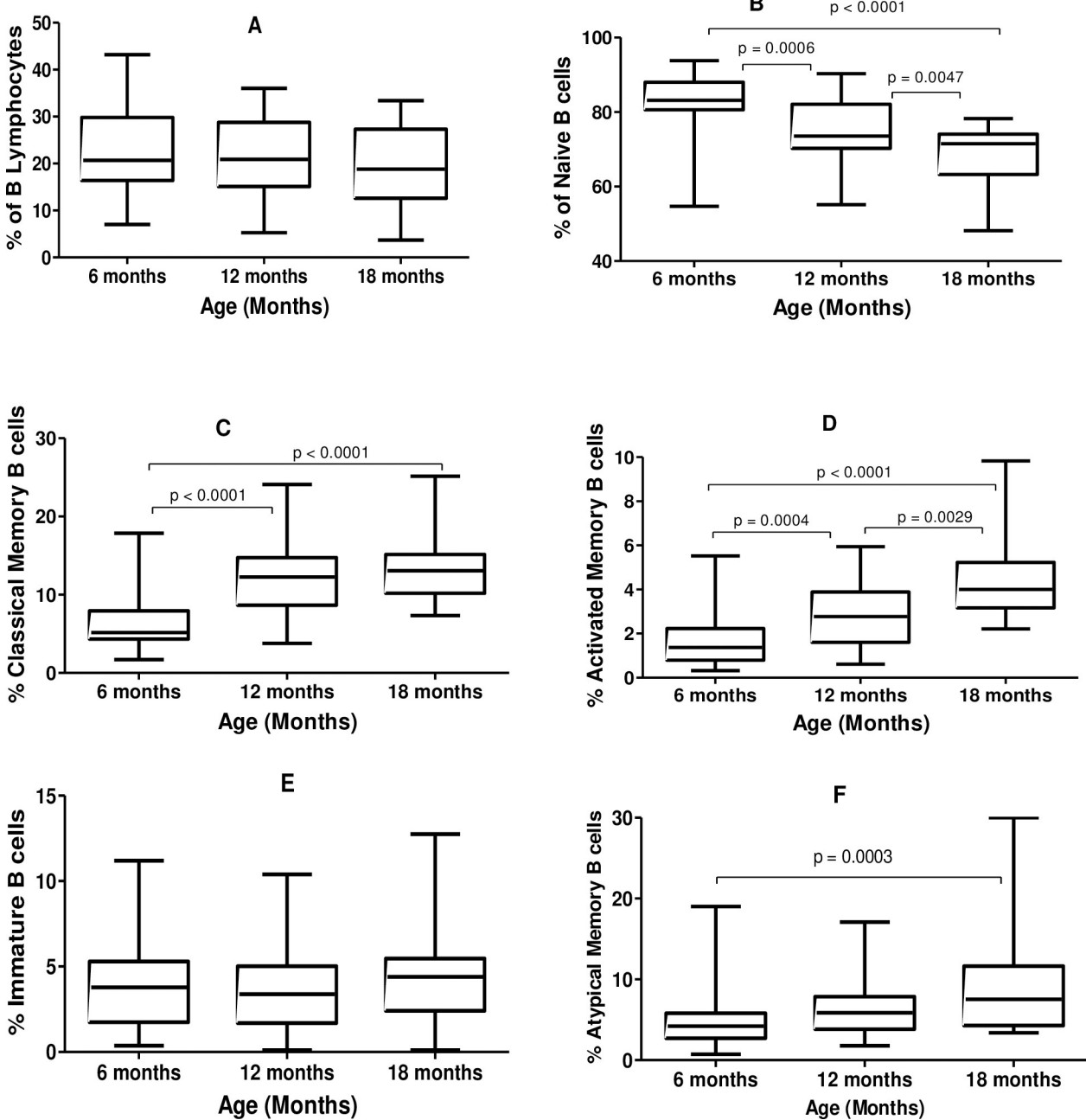

**Fig 2. Median proportions of naïve and memory CD19$^+$ B cell subsets in healthy children at different ages.** (A) Percentage of B lymphocytes, (B) proportions of naïve, (C) classical memory, (D) activated memory, (E) immature transitional and (F) atypical memory B cells.

significant differences in CD4+ and CD8+ T cells [26, 35]. Further analysis of the data for the first part of this study confirmed this observation for B cells (S2 Table in S1 File).

Although social habits such as smoking and dietary patterns can potentially affect the proportion of different lymphocyte subsets [3, 19], the inclusion and exclusion criteria for recruitment of study participants for this work were stringent enough to ensure that all study participants were non-smokers and were not in any way malnourished or overweight or presenting with any known disease at the time of recruitment.

**Table 4. Medians (10th and 90th percentiles) of proportions and absolute counts of B cells in study participants from different countries in the age range of 18 to 40 years.**

|  |  | % CD19+ B cells | CD19+ B cells/µl |
|---|---|---|---|
| 1 | Our study (n = 53) | 11.88 (7.51–18.23) | 253 (144–454) |
| 2 | Ethiopia [1] (n = 51) | 11.0 (4.0–24.0) | 181 (56–436) |
| 3 | Tanzania [17] (n = 214) | 13.0 (6.0–21.0) | 253 (88–654) |
| 4 | Holland [20] (n = 678) | NA | 290 (110–670) |
| 5 | Germany [21] (n = 100) | NA | 220 (80–490) |
| 6 | Germany [22] (n = 32) | 9.2 (7.2–11.2) | 199 (169–271) |
| 7 | USA [10] (n = 784) | NA | 300 (110–570) |
| 8 | Cuba [23] (n = 129) | 25.6 (5.4–49.5) | 452 (114–1,491) |
| 9 | China [24] (n = 135) | 13.00 (8.84–17.76) | 316 (203–476) |
| 10 | Singapore [25] (n = 38) | 12.0 (10.0–15.0) | 200 (100–200) |
| 11 | Korea [26] (n = 294) | 10.43 (10.0–10.86) | 203 (191–214) |
| 12 | Saudi Arabia [27] (n = 30) | 13.9 (5.0–27.0) | 298 (110–730) |

The main limitation of the first part of the study was that, due to the flow cytometer we had access to at the time (Four-colour BD FACS Calibur), we could only study the two main subsets of B cells (naïve and memory) but could not include the other subsets which we were able to study in the second cohort with the use of the Nine-Colour Cyan flow cytometer. The second cohort's main limitation was the reduced sample sizes and the limited follow up period of three time points (6, 12 and 18 months). The next obvious study to build on these results should aim at following up the children from 1 month to 60 months with several time points in between and analyse the blood samples for even more B cell subsets such as switched and non-switched memory B cells, CD27-negative memory B cells, transitional B cells as well as CD21lowCD38low B cells as some [22] have done.

## Conclusion

We have shown that B cell subsets in Malawians vary with age but not by gender. The Proportion and absolute counts of B cells reach a peak in the 7-to-12-months age group and were lowest in those aged 60 years and above. These observations could be useful in the interpretation of the natural variation in B cell subsets in otherwise healthy individuals and provide insight on how they respond to infection and vaccination in this setting.

## Supporting information

**S1 File. Supplementary S1-S3 Figs and Supplementary S1 and S2 Tables.**
(DOCX)

## Acknowledgments

We are grateful to the children and families for their participation in the study.

## Author Contributions

**Conceptualization:** Wilson L. Mandala.

**Formal analysis:** Wilson L. Mandala.

**Investigation:** Wilson L. Mandala, Herbert Longwe.

**Methodology:** Wilson L. Mandala, Herbert Longwe.

**Writing – review & editing:** Wilson L. Mandala, Herbert Longwe.

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
