## [Decision Letter · Decision Letter 0]

26 Apr 2021

PONE-D-21-07762

Variation of B cell Subsets in Healthy Malawins

PLOS ONE

Dear Dr. Mandala,

Thank you for submitting your manuscript to PLOS ONE. After careful consideration, we feel that it has merit but does not fully meet PLOS ONE’s publication criteria as it currently stands. Therefore, we invite you to submit a revised version of the manuscript that addresses the points raised during the review process.

We look forward to receiving your revised manuscript.

Kind regards,

Calogero Caruso, MD

Academic Editor

PLOS ONE

Additional Editor Comments:

As suggested by the referees, the paper must be extensively corrected before it can be considered for acceptance

Journal Requirements:

2. Please provide additional details regarding participant consent. In the ethics statement in the Methods and online submission information, please ensure that you have specified what type you obtained (for instance, written or verbal, and if verbal, how it was documented and witnessed). If your study included minors, state whether you obtained consent from parents or guardians.

3. Please provide a sample size and power calculation in the Methods, or discuss the reasons for not performing one before study initiation.

4. In your Methods section, please provide additional information about the participant recruitment method and the demographic details of your participants. Please ensure you have provided sufficient details to replicate the analyses such as: a) the recruitment date range (month and year).

5. Please ensure your Methods and reagents are described in sufficient detail for another researcher to reproduce the experiments described. Specifically, please provide the source and product number of any antibodies purchased for this study.

6. Please include your tables as part of your main manuscript and remove the individual files. Please note that supplementary tables should remain/ be uploaded) as separate "supporting information" files.

7. Thank you for stating the following in the Funding Section of your manuscript:

"The study of the first cohort was funded by a PhD fellowship from he Bill and Melinda Gates

Foundation whereas the second cohort study was funded by the Wellcome Trust Grant to the

Malawi Government under HRCSI."

"The funders had no role in study design, data collection and analysis, decision to publish or preparation of the manuscript"

8. Thank you for submitting the above manuscript to PLOS ONE. During our internal evaluation of the manuscript, we found significant text overlap between your submission and the following previously published works, some of which you are an author.

- https://bmcimmunol.biomedcentral.com/articles/10.1186/s12865-015-0115-y

Please revise the manuscript to rephrase the duplicated text, cite your sources, and provide details as to how the current manuscript advances on previous work. Please note that further consideration is dependent on the submission of a manuscript that addresses these concerns about the overlap in text with published work.

Reviewers' comments:

Reviewer's Responses to Questions

**Comments to the Author**

1. Is the manuscript technically sound, and do the data support the conclusions?

Reviewer #1: Partly

Reviewer #2: Yes

2. Has the statistical analysis been performed appropriately and rigorously? 

Reviewer #1: Yes

Reviewer #2: No

3. Have the authors made all data underlying the findings in their manuscript fully available?

Reviewer #1: Yes

Reviewer #2: Yes

4. Is the manuscript presented in an intelligible fashion and written in standard English?

Reviewer #1: No

Reviewer #2: Yes

5. Review Comments to the Author

Reviewer #1: In this paper Mandala WL et al analysed B cell subsets in Malawians, concluding that B cell subsets vary with age. These analysis provide new insight in the knowledge of B cells.

Although the topic is interesting, the manuscript should be improved with a better presentation of the results.

Addressing the following issues would significantly enhance its predicted appeal:

1) Results section needs careful review; some part of results are duplicated. The number of tables must also be revised and corrected. In the Results Figure 5 is not fully mentioned

2) Discussion is too short: authors should better describe the results obtained and their scientific value, even by updating the bibliography which is old.

3) Regarding the flow cytometry analysis have the authors used isotype control mAbs? It is necessary because some representative dot plot are not convincing.

4) Figure 5 is not mentioned in Methods, Results and Discussion.

5) Please, check the English style and the typo errors

My final assessment of this manuscript will depend on the revision of the specific points above reported.

Reviewer #2: In this study, Mandala and Longwe describe the changes in B cells occurring in a healthy cohort of subjects from Malawi. The study shows basically the same results observed in other cohorts from nearby nations. As correctly acknowledged by the authors, the technical limitations of the instrument did not allow to have a deep characterization of B cells in this samples, and this dampens the interest of the study.

There are some concerns and flaws that have to be addressed in the study:

1. A comparison with average values of other populations have to be reported and further discussed, maybe by adding a reference value in the tables proposed by the authors. Does this difference change with age?

2. Samples are not balanced for sex, and the M/F ratio changes significantly in different age groups. Does this influence the proportion of B cell subsets? I strongly suggest to show data after being stratified according to sex. This is also true for the longitudinal study focused on neonates.

3. How does the age of subjects >60 is distributed?

4. An appropriate statistic test (e.g. Kruskall Wallis test) should be used when comparing three or more groups, like in this case.

5. More details concerning health status or demographic features of enrolled subjects should be provided, maybe in a table.

5. In the discussion it would be worthy of interest the comparison between this cohort and other cohorts that include people with different lifestyle or environmental conditions, to evaluate the possible impact of these variables on B cells.

6. Figure 1 is useless, as very basic and standard, and could be moved to supplementary.

6. In the figure 2, there are some misalignments of arrow and legends

6. PLOS authors have the option to publish the peer review history of their article (what does this mean?). If published, this will include your full peer review and any attached files.

Reviewer #1: No

Reviewer #2: No

---

## [Author Response · Author response to Decision Letter 0]

7 Jun 2021

A. Academic Editor’s Comments

Our Response: We have now made these changes throughout the manuscript.

2. Please provide additional details regarding participant consent. In the ethics statement in the Methods and online submission information, please ensure that you have specified what type you obtained (for instance, written or verbal, and if verbal, how it was documented and witnessed). If your study included minors, state whether you obtained consent from parents or guardians.

Our Response: We have now provided these additional details on page 11 under Ethics Approval section of the revised version with track changes.

3. Please provide a sample size and power calculation in the Methods, or discuss the reasons for not performing one before study initiation.

Our Response: We have now included a paragraph on page 7 of the revised version in which we explain the sample size calculation basis for our study 

4. In your Methods section, please provide additional information about the participant recruitment method and the demographic details of your participants. Please ensure you have provided sufficient details to replicate the analyses such as: a) the recruitment date range (month and year).

Our Response: We have now provided these additional details on pages 6 and 7 under the Participants section of the revised version with track changes.

5. Please ensure your Methods and reagents are described in sufficient detail for another researcher to reproduce the experiments described. Specifically, please provide the source and product number of any antibodies purchased for this study.

Our Response: We have now expanded the section describing how the Immunophenotyping work was done for both the first and second part of the study. The revised section can be found on pages 8 and 9 under the section entitled Blood Samples Analyses and more details on the monoclonal antibodies are now provided in S1 Table.

6. Please include your tables as part of your main manuscript and remove the individual files. Please note that supplementary tables should remain/ be uploaded) as separate "supporting information" files.

Our Response: We have now included the main Tables as part of the main manuscript. However S1 Table, which has more details on the monoclonal antibodies in this study, is included in a separate file entitled “Supporting Information File”

7. Thank you for stating the following in the Funding Section of your manuscript:

"The study of the first cohort was funded by a PhD fellowship from the Bill and Melinda Gates Foundation whereas the second cohort study was funded by the Wellcome Trust Grant to the Malawi Government under HRCSI." We note that you have provided funding information that is not currently declared in your Funding Statement. However, funding information should not appear in the Acknowledgments section or other areas of your manuscript. We will only publish funding information present in the Funding Statement section of the online submission form.

"The funders had no role in study design, data collection and analysis, decision to publish or preparation of the manuscript"

Our Response: Thanks so much for bringing this point to our attention. We have now deleted the funding information which was originally included after the Acknowledgement Section. May you please change our Funding Statement to read:

"The study of the first cohort was funded by a PhD fellowship for Wilson Mandala from the Bill and Melinda Gates Foundation under the Gates Malaria Partnership (GMP) for African countries whereas the second cohort study was funded by the Wellcome Trust Grant to the Malawi Government under the Health Research Capacity Strengthening Initiative (HRCSI) for which both Wilson Mandala and Herbert Longwe were beneficiaries"

8. Thank you for submitting the above manuscript to PLOS ONE. During our internal evaluation of the manuscript, we found significant text overlap between your submission and the following previously published works, some of which you are an author.

- https://bmcimmunol.biomedcentral.com/articles/10.1186/s12865-015-0115-y

Please revise the manuscript to rephrase the duplicated text, cite your sources, and provide details as to how the current manuscript advances on previous work. Please note that further consideration is dependent on the submission of a manuscript that addresses these concerns about the overlap in text with published work.

Our Response: Thanks so much for bringing this point to our attention. Indeed there were some paragraphs in the Introduction and Materials and Methods section that sounded very similar to those found in our earlier publication (Longwe et al, BMC Immunol 2015). We have now revisited these paragraphs and edited them accordingly on pages 4 and 9.

B. First Reviewer’s Comments

In this paper Mandala WL et al analysed B cell subsets in Malawians, concluding that B cell subsets vary with age. These analysis provide new insight in the knowledge of B cells.

Although the topic is interesting, the manuscript should be improved with a better presentation of the results. Addressing the following issues would significantly enhance its predicted appeal:

1. Results section needs careful review; some parts of results are duplicated. 

Our Response: We thank the reviewer for bringing this observation to our attention. We have now revised the entire Results section and deleted any duplicated segments.

2. The number of tables must also be revised and corrected. In the Results Figure 5 is not fully mentioned

Our Response: We thank the reviewer for bringing this observation to our attention. We have now revised the numbering of the Tables and Figures. In addition we have now referred to what was previously Figure 5 but is now S3 Figure on Page 6 in the Materials and Methods section and on page 14 under the Results section.

3. Discussion is too short: authors should better describe the results obtained and their scientific value, even by updating the bibliography which is old.

Our Response: We have now revisited the entire Discussion section and included more recent reports in our Reference section

4. Regarding the flow cytometry analysis have the authors used isotype control mAbs? It is necessary because some representative dot plot are not convincing.

Our Response: We have expanded the section on Blood Sample Analyses in the Materials and Methods section where we do explain that we had included isotype controls in our analyses (Pages 8 and 9). We have also provided the details of all monoclonal antibodies used in our study and these are provided in S1 Table.

5. Figure 5 is not mentioned in Methods, Results and Discussion.

Our Response: We have now referred to what was previously Figure 5 but is now S3 Figure on Page 6 in the Materials and Methods section and on page 14 under the Results section.

6. Please, check the English style and the typo errors

Our Response: We have gone through the entire manuscript and cleaned up all typo errors and English styles,

My final assessment of this manuscript will depend on the revision of the specific points above reported.

C. Second Reviewer’s Comments

In this study, Mandala and Longwe describe the changes in B cells occurring in a healthy cohort of subjects from Malawi. The study shows basically the same results observed in other cohorts from nearby nations. As correctly acknowledged by the authors, the technical limitations of the instrument did not allow to have a deep characterization of B cells in this samples, and this dampens the interest of the study.

There are some concerns and flaws that have to be addressed in the study:

1. A comparison with average values of other populations have to be reported and further discussed, maybe by adding a reference value in the tables proposed by the authors. Does this difference change with age?

Our Response: We have now added Table 4 in the Discussion section (page 17) which provides Medians and interquartile ranges of the proportions and absolute cell numbers of B cells that have been reported in various studies done in different countries and for different ethnicities. We have referred to these different values in the Discussion section.

2. Samples are not balanced for sex, and the M/F ratio changes significantly in different age groups. Does this influence the proportion of B cell subsets? I strongly suggest to show data after being stratified according to sex. This is also true for the longitudinal study focused on neonates.

Our Response: Indeed we agree with the Reviewer that the male: female ratio changes between groups much as we tried to balance these during recruitment time. We have now re-analysed the data for the first part of the study by sex and included S2 Table which shows that there were NO significant differences between the two sexes. We explain this in the manuscript on page 19 in the Discussion section.

3. How does the age of subjects >60 is distributed?

Our Response: We have now provided a paragraph on page 12 explaining how the study participants aged 60 years and above were distributed. In short we found that the 55 participants in the group aged 60 years or more had a media age of 66 years with the age ranging from 60 to 92 years.

4. An appropriate statistic test (e.g. Kruskall Wallis test) should be used when comparing three or more groups, like in this case.

Our Response: We have had to revisit our statistical analysis and used Kruskal Wallis test instead of Mann Whitney as we had previously done. In addition, we have used Dunn’s Multiple Comparison test and this is explained on pages 9 and 10

5. More details concerning health status or demographic features of enrolled subjects should be provided, maybe in a table.

Our Response: We have now added these details on pages 6 and 7

6. In the discussion it would be worthy of interest the comparison between this cohort and other cohorts that include people with different lifestyle or environmental conditions, to evaluate the possible impact of these variables on B cells.

Our Response: We have done this in the Discussion section on page 19.

7. Figure 1 is useless, as very basic and standard, and could be moved to supplementary.

Our Response: We have moved what was Figure 1 to become S1 Figure.

8. In the figure 2, there are some misalignments of arrow and legends

Our Response: We have moved what was Figure 2 to become Supplementary Figure 2 (S2) and have edited the Figure accordingly to address any misalignments that were originally there.

---

## [Decision Letter · Decision Letter 1]

24 Jun 2021

Variation of B cell Subsets in Healthy Malawians

PONE-D-21-07762R1

Dear Dr. Mandala,

We’re pleased to inform you that your manuscript has been judged scientifically suitable for publication and will be formally accepted for publication once it meets all outstanding technical requirements.

Kind regards,

Calogero Caruso, MD

Academic Editor

PLOS ONE

Additional Editor Comments (optional):

Reviewers' comments:

Reviewer's Responses to Questions

**Comments to the Author**

1. If the authors have adequately addressed your comments raised in a previous round of review and you feel that this manuscript is now acceptable for publication, you may indicate that here to bypass the “Comments to the Author” section, enter your conflict of interest statement in the “Confidential to Editor” section, and submit your "Accept" recommendation.

Reviewer #1: All comments have been addressed

Reviewer #2: All comments have been addressed

2. Is the manuscript technically sound, and do the data support the conclusions?

Reviewer #1: Yes

Reviewer #2: Yes

3. Has the statistical analysis been performed appropriately and rigorously? 

Reviewer #1: Yes

Reviewer #2: Yes

4. Have the authors made all data underlying the findings in their manuscript fully available?

Reviewer #1: Yes

Reviewer #2: Yes

5. Is the manuscript presented in an intelligible fashion and written in standard English?

Reviewer #1: Yes

Reviewer #2: Yes

6. Review Comments to the Author

Reviewer #1: The authors have exhaustively improved the quality and the significance of their results. In the new form, the manuscript is acceptable for the publication

Reviewer #2: The authors have addressed my concerns in a satisfactory manner. I have no further comments or requests.

7. PLOS authors have the option to publish the peer review history of their article (what does this mean?). If published, this will include your full peer review and any attached files.

Reviewer #1: **Yes: **Meraviglia Serena

Reviewer #2: **Yes: **Marcello Pinti

---

## [Editor Report · Acceptance letter]

30 Jun 2021

PONE-D-21-07762R1 

Variation of B cell Subsets with age in Healthy Malawians 

Dear Dr. Mandala:

I'm pleased to inform you that your manuscript has been deemed suitable for publication in PLOS ONE. Congratulations! Your manuscript is now with our production department. 

Kind regards, 

on behalf of

Prof. Calogero Caruso 

Academic Editor

PLOS ONE